# Increased awareness and decreased acceptance of genome-editing technology: The impact of the Chinese twin babies

**Daiki Watanabe**[1], **Yoko Saito**[1]*, **Mai Tsuda**[2], **Ryo Ohsawa**[2]

**1** Research Faculty of Agriculture, Hokkaido University, Sapporo, Hokkaido, Japan, **2** Faculty of Life and Environmental Sciences, University of Tsukuba, Tsukuba, Ibaraki, Japan

* saitoy@agecon.agr.hokudai.ac.jp

**Data Availability Statement:** All relevant data are available within the paper and supporting information files.

## Abstract

Genome-editing technology has become increasingly known in recent years, and the 2018 news of genome-edited twins in China had a particularly significant impact on public awareness. In the present study we investigate the effect of Japanese media coverage on public opinions of this technology. To identify the effects we employ a questionnaire survey method on a pre-registered sample from online research company Macromill. Our repeated survey from 2016 through 2019 reveal a generally supportive attitude toward the medical application of genome-editing methods. To see this we employed a multinomial logit analysis examining the determinants of negative and positive impressions of the technology. Results show that although editing for medical purposes remained mostly acceptable, its use in fertilizing human eggs was increasingly rejected, especially in 2019, the most recent sample year. The suggestion is that while genome-editing applications in general medical fields are publicly accepted, its use in human functionality enhancement is heavily increasingly resisted. News of the twin babies in China did raise public awareness of the methods but also damaged their reputation. It therefore is important for genome researchers to hold such concerns in mind, keeping the public informed of changing technology fundamentals. As a related question we inquire into the public acceptability of genome editing for animal and plant breeding, such as in agriculture and fisheries, as well. We find the Japanese public views the medical and breeding applications of this technology to be unconnected with each other, despite that awareness of both has risen significantly in recent years.

## Introduction

The emerging technology of genome editing, together with its applications, is attracting increased public attention, especially in its use with human genes. Many in the scientific community are calling for debate on the appropriate applications of this technology, one that will draw scientists, bioethicists, legal and regulatory bodies, and the public together [1–3]. Since these calls, the public's perception and acceptance of human gene editing are starting to be investigated. Discussion focuses mainly on the technology's medical applications, such as

**Funding:** This research was supported (in part) by Cooperative Research Grant # 2043 of the Plant Transgenic Design Initiative (PTraD) by Gene Research Center, Tsukuba-Plant Innovation Research Center, University of Tsukuba, and a grant from the Cross-ministerial Strategic Innovation Promotion Program (SIP), Technologies for creating next-generation agriculture, forestry and fisheries by the Bio-oriented Technology Research Advancement Institution, NARO.

**Competing interests:** The authors have declared that no competing interests exist.

therapeutic or function enhancement purposes–and as an extension, the moral concerns of both the scientific and general community [4–9].

An earlier study [5] collected 12,000 respondents through social media, finding public acceptability of the technology for therapeutic uses differed starkly from that for reproductive uses. Another survey indicated more than 60% of respondents favored the technology's application in medical treatments but disfavored its use in functionality enhancements, such as applications to human embryos [4]. And more knowledgeable respondents indicate the need for science community to consult with public. Another survey [6], based on respondents from eleven countries including the US and European nations, suggested medical applications' functionality enhancement purposes are not more acceptable than therapeutic purposes are. In Australia, respondents were accepting of human gene editing for both research and human health purposes [7].

Surveys of the Japanese public have found similar results, and begun to go further in asking how one's understanding or 'literacy' of genetic editing affect its acceptability. The more gene-literate respondents seem to show a more positive attitude toward its medical applications [10], though literacy has no significant effect on attitudes toward crop applications [11]. Survey [12] found gene-literate respondents to be even more cautious about gene manipulation than the less literate is. Disease-abatement applications are more acceptable than others, even when the respondent's knowledge of the technology is small [13]. And the literature [14] suggests an ambiguity in the Japanese public toward genome editing methods in general. Media coverage of the birth of genetically edited twins in China included a great deal of ancillary information about the method's applications in medical science. Media exposure doesn't always alleviate resistance to biotechnology, it is found [15], and the impact of the twins' birth in particular has not yet been examined.

In the present study, public acceptance of genome-editing technology was investigated by way of a sequence of surveys conducted in 2016, 2018, and 2019, with a focus on media impacts. Genome-editing methods have become widely known in Japan in recent years, and public attitudes are changing accordingly. Fig 1 shows that from 2015 to 2019 the search-volume frequency of the term "genome-editing technology" peaked in Japan during the week of November 26, 2018 when news of the twin babies was released in China [16]. News reports

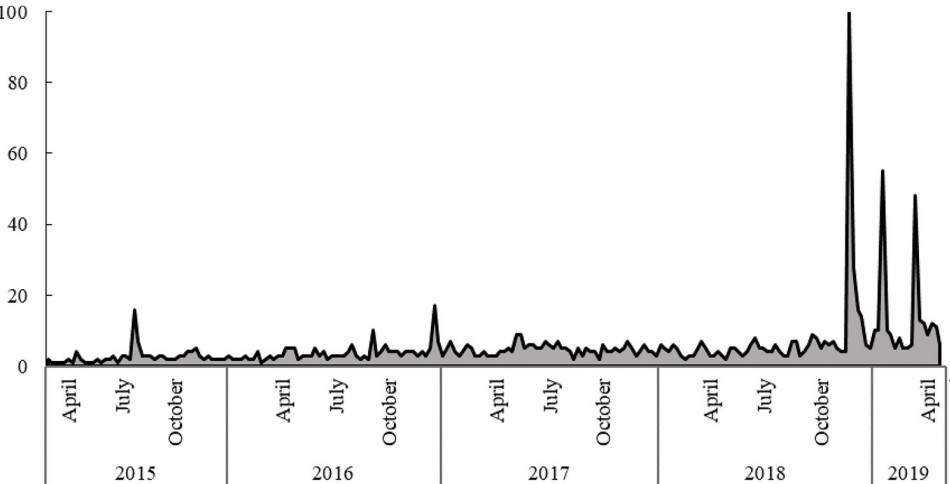

**Fig 1. Search volume of term "genome-editing technology" in Japanese (Google).** The greatest number of searches recorded (100) was in the week of November 26, 2018.

detailed the manner in which genome-editing technology was applied to a human embryo, and it was widely criticized in both academic and non-academic groups. The coinciding of the search-volume peak in late 2018 (Fig 1) seems to suggest that the controversy added substantially to public awareness of genome-editing methods. However, this has not been examined carefully until now, and it is especially important to do so during this early stage of the technology's diffusion.

Media coverage can be an influencing factor in consumer preferences, as revealed for example in bovine spongiform encephalopathy (BSE), E. coli O157 cases [17], and kepone contamination [18]. These studies show media health data can discourage market demands for specific foods. Here we investigate whether it also affects the demand for genome editing technology. It is important to clarify that no consumer products from genome-editing technology have yet been marketed, so it was impossible for us to trace demand changes with market data. Survey questionnaires were thus used instead, as they are the most suitable alternatives to market information.

Biotechnology's medical uses generally receive strong public support [19]. But even in these uses, acceptance likely will differ according to the technology's purposes and to an individual's knowledge or literacy of genetics. Given that both academics and the public received news of the Chinese twins with particular disfavor, we hypothesize that the acceptability of *genome-editing technology itself* was negatively affected by it. The goal of our study is to investigate the degree to which public acceptance would be affected by media coverage in general, and whether the degree of that influence would depend on the purposes of its use and on one's knowledge of the methods. We thus separately specify a therapeutic use and a functionality-enhancement use of the technology to enable us to identify any differences in news-reader impressions that they might evoke.

We also analyzed the impact of news on genome editing applications in areas such as agriculture and fishery breeding. Although the main reason for the public's response to the Chinese twins is that it was applied to a human embryo, it may also have affected sensitivities to applications in other areas as well. Genome editing, one of the new plant-breeding technologies (NBTs), can induce mutations more precisely than earlier methods could [20]. At the same time, they can be understood as a method similar to conventional cross-breeding. Rapid progress in molecular biology has facilitated progressively wider plant gene modifications [21–22]. Yet even there, gene recombinant technology in particular has raised dietary worries [23]. Thus, although genome editing technology as a whole is considered scientifically promising, attitudes toward its various applications were perhaps entangled with one another when the term first became publicly known.

## Survey and results

### Survey description

Our survey of the public awareness/acceptance of genome-editing technology and related media coverage was conducted in Japan in March 2016, January 2018, and finally January 2019 (Table 1). For each survey, respondents were independently selected out of a pre-registered sample from the online survey company Macromill. The numbers of pre-registered respondents at hand to us at the times of our survey is shown in Table 1, and the associated summary statistics displayed in S1 Table. Original monitors in the pre-registered sample showed that 18.0% of respondents were under 19 years old and were excluded from our survey. According to Macromill, pre-registered samples are generated by voluntary registration.

As our survey was conducted by the research firm Macromill, the sample had already been anonymized. We were allowed to access only to the respondents' ID numbers, which are

**Table 1. Survey parameters.**

|  | 2016 | 2018 | 2019 |
|---|---|---|---|
| Survey period | March 2016 | Jan-Feb 2018 | Jan-Feb 2019 |
| Questionnaire title | Survey related to science and technology | Food Questions | Food Questions |
| Number of respondents[a] | 3,100 (682) | 1,240 (422) | 1,543[b] (677) |
| Number of respondents in the pre-registered sample (in thousands) | 901.2 | 1,196.1 | 1,215.8 |
| Note on surveyed sample | Between 20–60 year age groups, equal numbers of males and females | Between 20–60 year age groups, equal numbers of males and females | An unequal number of males and females[c] |

[a] Numbers in parentheses are those who replied "very knowledgeable," "know some," or "have heard" about genome-editing technology in Q3. Any who replied "Do not remember" in the following Q4 were also excluded. In each category, five alternatives were offered (S1 Text): "nothing at all," "not much," "have heard of it," "have some information," and "very knowledgeable."

[b] Because seven respondents had been sampled in both the 2016 and 2019 surveys, they were excluded from the 2019 sample.

[C] The number of females and males in the 2019 survey were unequal because of the exclusion (see footnote b).

individually assigned upon their registration with Macromill's pre-registered pool. Respondents hold the right to withdraw from or refuse to participate in the survey at any time during the online survey procedure. Given these provider rights and anonymity, written informed consent was not required. In addition to this however, the Research Ethics Committee of the Research Faculty of Agriculture, Hokkaido University, has formally confirmed our compliance.

Respondents to our survey were solicited on the basis of the questionnaire title sent from the survey company or posted on the company website, which remained open until the number of survey responses reached our requested sample size. We requested a sample with an equal number of male and female respondents aged between their 20s and 60s. The total number of respondents was 3,100 in 2016, 1,240 in 2018, and 1,550 in 2019. The pre-registered sample was 901,198 in 2016, 1,196,092 in 2018, and 1,215,789 in 2019. As we requested an equal number of respondents by sex and age group, the shares of these groups in our sample are largely the same across groups. We were not able to check for any statistically significant differences between the pre-registered samples and our surveyed ones, since standard deviations in the pre-registered sample are unavailable. But location and occupation shares are quite similar in the two samples.

Each year's sample excluded those who had replied the previous year. To reduce opinion bias, we avoided using "genome-editing technology" in the questionnaire title when soliciting respondents from the pre-registration sample, and instead used more general terms like "survey on food" and "survey on science and technology." Survey sheets in both the original language (Japanese) and the English translation are shown in the Questionnaire in S1 Text.

## Content of media coverage

We asked respondents to indicate the degree of their awareness of genome-editing technology on a five-point Likert scale. We then, for some purposes, grouped those who replied "very knowledgeable," "know some information," or "have heard of it" as broadly 'being aware' of genome-editing methods (Fig 2 and S2 Table). Results show the proportion of those aware of the technology in this sense has been rising, from 28.8% in 2016 to 51.8% in 2019, implying the term "genome-editing technology" has become increasingly familiar in the past few years.

Subsequent analysis was undertaken only with the respondents who were aware of genome technology according to that division, since those who aren't would be unable to answer several of the important questions, such as whether they had encountered any media coverage on

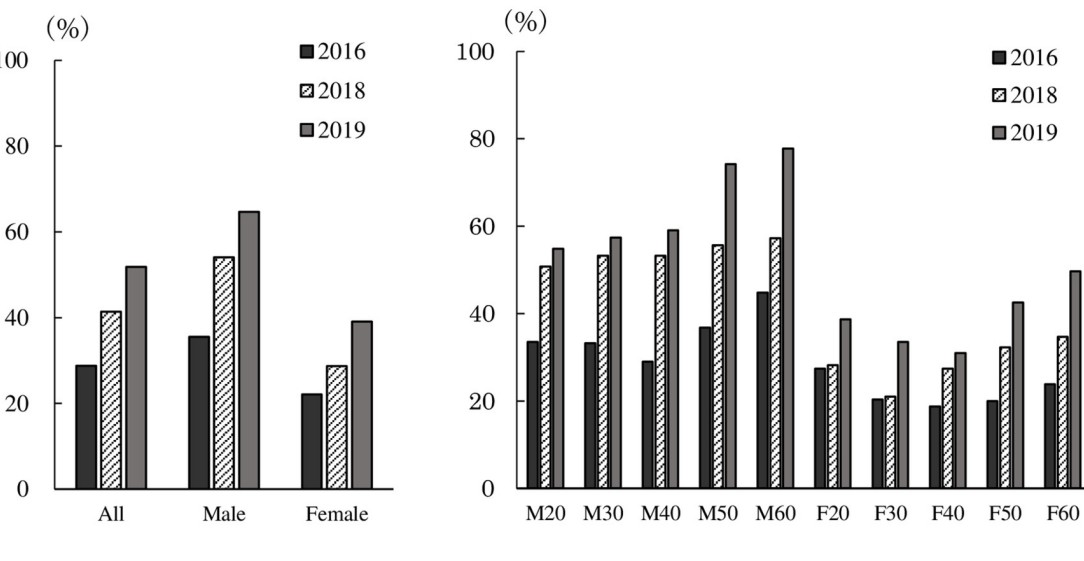

(a) Proportion by year and sex                    (b) Proportion by year and age groups

**Fig 2. Proportion of those who "have heard", "know some" or are "very knowledgeable" of the term "genome-editing technology," shown by survey year, and further divided by (a) sex, and (b) age group.**

the subject. Summary statistics of both "not being aware" and "being aware" sub-samples are provided in S3 Table. Our results are consequently drawn from respondents who showed at least some interest in the technology. This may have an upward effect on the numbers of those who have a positive attitude. We could have provided in the online survey additional objective information, but respondents' own views of a technology likely depend on the contexts in which the technology will be used, which inherently are not objective. Respondents saying they were unaware or 'not much' aware of gene editing were thus encouraged to skip these particular questions.

We further asked respondents who were aware of genome-editing technology which media content made the strongest impression (positive or negative) on them (Table 2). A number of possible media categories were suggested in this regard, including "an explanation of a technology," "an application to fishery or agricultural breeding," "an explanation of the risk of technology adoption," "medical applications of the technology," and "applications to fertilized human eggs."

An "explanation of the technology itself" created the strongest impression (38.9%) in 2016, whereas "medical application" created the strongest in 2018 (32.2%) and 2019 (28.5%). Respondents on whom use in fertilized human eggs made the strongest impression rose from 15.9% in 2018 to 25.4% in 2019, suggesting the news of the twin babies in China had a substantial influence on the Japanese public.

The variety of types of impressions created by the media coverage of genome editing is shown in Table 3. The number of respondents saying they found it to be "interesting or impressive" was the highest response in 2016 and 2018, while the number saying they were "concerned about its side or unknown effects" was highest in 2019. As public awareness rose, safety issues, including uncertainties over the method's side effects, began to receive the greater attention. Concerns about its ethical status were quite high in the 2018 and 2019 surveys, consistent with the increasing attention being paid to research ethics.

**Table 2. Content making the strongest impression [a].**

| Media Content | 2016 | 2018 | 2019 |
|---|---|---|---|
| Explanation of the technology itself | **265** | 81 | 112 |
| (*c_tech*) | **(38.9)** | (19.2) | (16.5) |
| Application to fishery or agricultural breeding technology | 210 | 101 | 137 |
| (*c_breed*) | (30.8) | (23.9) | (20.2) |
| Explanation of the risk of adopting the technology | 76 | 31 | 54 |
| (*c_risk*) | (11.1) | (7.4) | (8.0) |
| Medical application of the technology | 125 | **136** | **193** |
| (*c_medical*) | (18.3) | **(32.2)** | **(28.5)** |
| Application to fertilized human eggs | - | 67 | **172** |
| (*c_embryo*) | - | (15.9) | **(25.4)** |
| Other Media Categories | 6 | 6 | 9 |
| (*c_others*) | (0.9) | (1.4) | (1.3) |
| Total | 682 | 422 | 677 |

[a] A respondent could select only one media content. Tabled values indicate the number of respondents choosing that category. Parenthesized numbers are the corresponding percentage shares of the sampled respondents.

## Analytical framework

We selected a subset (see asterisked responses) of the impression responses in Table 3 most suitable for characterizing, respectively, an overall supportive and unsupportive attitude toward the technology. In particular we classified, as being "positive" about the technology, those who gave "look forward to its adoption" a higher Likert mark than to "a little worried."

**Table 3. Most closely matched impressions (or evaluation of each of the impressions).**

| | 2016[a] | 2018[b] | 2019[b] |
|---|---|---|---|
| | Number Saying "Most Impressionable" (%) | Share of Those Agreeing with the Following: (Ave. score) | Share of Those Agreeing with the Following: (Ave. score) |
| Interesting or impressive | **269 (39.4)** | **83.4 (3.03)** | 78.7 (2.97) |
| Amazed by this technological advancement | 191 (28.0) | 79.9 (2.97) | 81.4 (3.02) |
| *Looking forward to its adoption | 49 (7.2) | 70.1 (2.87) | 64.8 (2.76) |
| *Worried about its adoption | 84 (12.3) | 64.0 (2.77) | 74.3 (2.94) |
| Concerned about its side or unknown effects | 62 (9.1) | 72.0 (2.89) | **82.1 (3.06)** |
| Concerned about ethical issues | (None) | 71.8 (2.89) | **79.6 (3.06)** |
| Do not understand the reason for its adoption | 4 (0.6) | 24.4 (2.11) | 37.8 (2.32) |
| Complicated explanation | 22 (3.2) | 34.6 (2.30) | 42.7 (2.40) |
| Other | 1 (0.2) | 0.9 (- -) | 1.2 (- -) |

[a] In 2016 we asked respondents to rate the top three of these impression types. This table shows the number of those marking the indicated impression type as that which "most closely matches my own impression." The number in parenthesis is that impression type's percentage share of those rated as 'top'.

[b] In 2018 and 2019 in contrast, we asked respondents to evaluate *each* impression on a four-point Likert scale: "Strongly agree (4)," "Somewhat agree (3)," "Disagree (2)," and "Strongly disagree (1)." The table indicates the number of respondents who marked that impression type as a (4) or a (3). The corresponding number in parenthesis is that impression type's percentage share of those receiving a (4) or a (3).

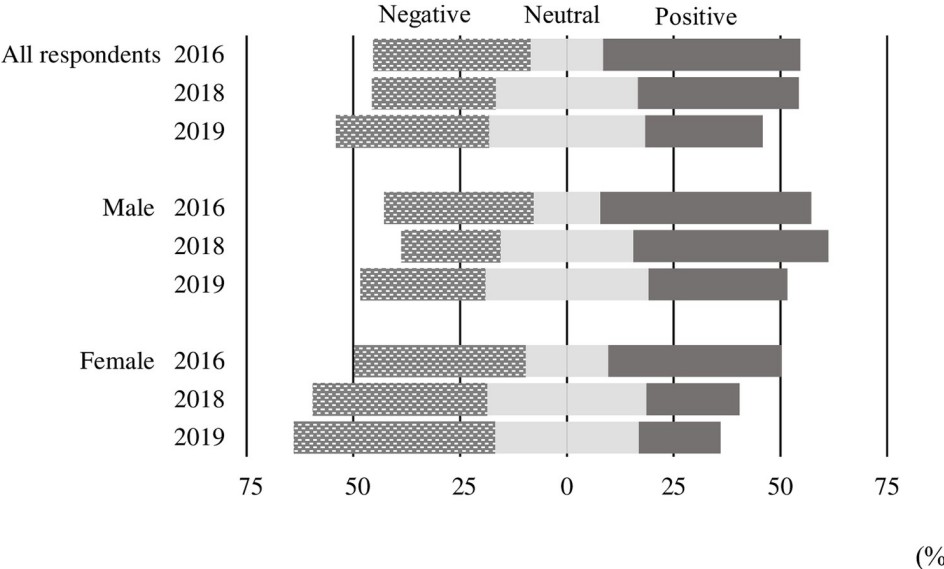

**Fig 3. Impressions of genome-editing technology.** Definitions of impressions are calculated from Table 3, specifically comparing the rate between impressions "Looking forward to its adoption" and "Worried about its adoption". Survey results are provided in Table B of S2. The present figure is drawn based on the share of each impression in S2 Table. Although the length of the bar is same, composition of impressions differ by year and sex. We set neutral to be center.

Those who gave a lower mark to it were classified as being "negative." Those giving the two an equal Likert score were "neutral," that is, were considered to be indifferent or undecided toward this new technology. Results of these comparisons are shown in Fig 3 and S2 Table.

As shown in Fig 3, as "genome-editing technology" became more widely known, both male and female attitudes tended to shift from positive to either negative or neutral. Considering all these respondents, positive attitudes fell from 46.2% in 2016 to 27.6% in 2019 (S2 Table), and the share of those with a "neutral" or "negative" attitude rose.

### Effect of media coverage on respondents' impressions

**Multinomial logit.** We have investigated how public impressions of genome-editing technology appeared to change in the period between 2016 and 2019 as public awareness of the technology grew. We now look to the factors influencing these impressions, using the multinomial logit approach often employed in the analysis of consumer decision-making. Previous research [24] has used this approach to demonstrate the gap between consumer and producer perceptions of meat risk. Here we employ a similar approach to quantify the effect of media coverage and personal attributes on public support for gene editing.

Our model specification is the following. The respondent's evaluation $Y_{ji}$ of the technology, in which $j$ is the evaluation category and $i$ is the respondent, is determined by the media contents ($X_{1i}$) he or she finds important, and by the respondent's characteristics ($X_{2i}$):

$$Y_{ji} = f(X_{1i}, X_{2i}) \tag{1}$$

where a possible evaluation category is "positive," "negative," or "neutral." That is, as detailed

further in Fig 3:

$$j = \begin{cases} 1 : neutral \\ 2 : negative \\ 3 : positive \end{cases}$$

and

$X_{1i}$: Vector of the media content types to which respondent $i$ responds:

($c\_tech$, $c\_breed$, $c\_risk$, $c\_medical$, $c\_embryo$, $c\_others$)

$X_{2i}$: Vector of respondent $i$'s individual characteristics.

The following multinomial logit model was then estimated by maximum likelihood:

$$P(y_i = j | x_i) = \frac{exp(x'_i \beta_j)}{\sum_{j=1} exp(x'_i \beta_j)} \tag{2}$$

$$Y_{ji} = \beta_{0j} + \beta_{1j} X_{1i} + \beta_{2j} X_{2i} + u_{ij} \tag{3}$$

where $P$ in Eq (2) is the occurrence probability; $y_i$ is the outcome respondent $i$ chooses; vector $\beta$ in Eq (3) is the set of parameters to be estimated and comprising the impact of the given explanatory variable on the respondent's impression; and $u_{ij}$ is the error term. Put differently, $Y_{ji}$ indicates the impression $j$ of respondent $i$ as determined by media content ($X_{1i}$) and respondent $i$'s characteristics ($X_{2i}$). A positive (negative) parameter indicates that the corresponding explanatory variable increases (decreases) the probability of the given impression.

Multinomial logit Eq (3) were estimated by maximum likelihood, using "neutral to the technology" as the base category ($j = 1$). The parameters thus are interpreted as changes in the probability of the corresponding variable relative to the 'neutral' base group. Estimates were derived separately for each of the three survey years. Media-content variables ($X_{1i}$) are zero/one, in which "technical explanation ($c\_tech$)" was used as the base. Details of media contents ($X_{1i}$) and individual characteristics ($X_{2i}$) are shown in S1 Text, and descriptive statistics and data are presented in S4 and S5 Tables. Estimation results are shown in Tables 4–6.

## Discussion of marginal effects

Consider now the multinomial logit results in Tables 4–6. Respondents exposed to media coverage of the technology's medical applications ($c\_medical$) showed significantly favorable impressions to it in both 2018 and 2019, indicating a supportive attitude toward the medical use of genome-editing methods. However, the *marginal effect* of medical content on this favorable impression fell from 26.2% in 2018 to 13.5% in 2019. At the same time, the November 2018 revelation about fertilized human eggs ($c\_embryo$) displayed a strong negative response, especially of course in 2019, right after the China news was received. The corresponding marginal effect on that unfavorability was 12.0% in 2018, rising to 29.0% in 2019. Moreover, the China news brought the greatest change, relative to other types of media content, in the probability of having a negative attitude toward gene editing.

Together these findings are indicative of the substantial effect in Japan of the Chinese twins news on the public acceptability of gene editing, a finding similar to a study of the US reaction [4]. While publics appear to support the medical uses of genome editing in general, its use in functionality enhancement, as in fertilized human eggs, impairs that acceptability. Given that ethical concerns in our 2019 sample had become slightly stronger than they were the previous year (Table 3), compliance with ethical principles will be an important factor in public support.

**Table 4. Estimated parameters and marginal effects, 2016 [a,b].**

| | 2016 | | | |
| --- | --- | --- | --- | --- |
| | Negative | | Positive | |
| | Parameter | Marginal Effect | Parameter | Marginal Effect |
| c_breed | -0.178 | -0.026 | -0.082 | 0.009 |
| | (0.278) | (0.044) | (0.260) | (0.044) |
| c_risk | 0.717* | 0.273*** | -0.716* | -0.285*** |
| | (0.385) | (0.058) | (0.427) | (0.070) |
| c_medical | 0.009 | 0.029 | -0.169 | -0.041 |
| | (0.322) | (0.051) | (0.310) | (0.052) |
| c_embryo | - | - | - | - |
| | - | - | - | - |
| c_others | -0.330 | 0.182 | -1.602 | -0.328 |
| | (0.947) | (0.213) | (1.251) | (0.264) |
| age | 0.007 | 0.001 | 0.001 | -0.001 |
| | (0.008) | (0.001) | (0.008) | (0.001) |
| d_male | 0.002 | -0.056 | 0.351 | 0.083** |
| | (0.235) | (0.038) | (0.228) | (0.039) |
| d_univ | 0.127 | -0.002 | 0.190 | 0.025 |
| | (0.251) | (0.040) | (0.242) | (0.042) |
| d_bio | -0.282 | -0.038 | -0.156 | 0.008 |
| | (0.296) | (0.045) | (0.289) | (0.048) |
| breed_k | -0.004 | -0.004 | 0.019 | 0.005 |
| | (0.053) | (0.008) | (0.051) | (0.009) |
| social_aware | - | - | - | - |
| | - | - | - | - |
| tech_aware | - | - | - | - |
| | - | - | - | - |
| cons | 0.562 | - | 0.745 | - |
| | (0.557) | - | (0.541) | - |
| Log likelihood | -679.96 | | | |
| Log likelihood (const) | -699.71 | | | |
| Observation | 682 | | | |

[a] Standard errors are shown in parentheses.

[b] *, **, and *** indicate statistical significance at 10%, 5%, and 1%, respectively.

Crucially, the *risks* of genome editing, apart from any *hopes* or *expectations* of its effects, weigh heavily on public opinion.

Our model parameter pointing to the media coverage of the technology's risks, ($c\_risk$), especially in its *negative* impression, was highly significant, with one of the highest marginal effects on public unacceptability of any we modeled. In contrast, media coverage of genomics' uses in agriculture and fish breeding ($c\_breed$) appear conducive to *positive* impressions of it in 2018 and 2019, although in 2019 its parameter was significantly negative as well as positive. News of the Chinese twins did not affect the positive views toward genomic breeding. The public's views of medical versus farm and fish gene editing don't seem, that is, to have been confused with one another, although each raised awareness of the technology in general.

On the side of the respondents' own characteristics ($X_{2i}$), knowledge of breeding technology ($breed\_k$) significantly reduced the likelihood of an unfavorable view of gene-editing methods,

**Table 5. Estimated parameters and marginal effects, 2018 [a,b].**

| | 2018 | | | |
| --- | --- | --- | --- | --- |
| | Negative | | Positive | |
| | Parameter | Marginal Effect | Parameter | Marginal Effect |
| c_breed | 0.312 | -0.014 | 0.852** | 0.144** |
| | (0.387) | (0.063) | (0.380) | (0.068) |
| c_risk | 1.699*** | 0.248*** | 0.661 | -0.007 |
| | (0.558) | (0.080) | (0.658) | (0.111) |
| c_medical | -0.201 | -0.137** | 1.236*** | 0.262*** |
| | (0.393) | (0.063) | (0.358) | (0.062) |
| c_embryo | 0.872** | 0.120* | 0.432 | 0.015 |
| | (0.409) | (0.065) | (0.444) | (0.079) |
| c_others | 0.121 | 0.082 | -0.744 | -0.158 |
| | (0.982) | (0.178) | (1.243) | (0.239) |
| age | -0.004 | 0.001 | -0.020** | -0.004** |
| | (0.009) | (0.001) | (0.009) | (0.002) |
| d_male | -0.417 | -0.150*** | 0.933*** | 0.219*** |
| | (0.275) | (0.042) | (0.287) | (0.048) |
| d_univ | 0.153 | 0.027 | 0.007 | -0.011 |
| | (0.289) | (0.046) | (0.271) | (0.049) |
| d_bio | -0.072 | -0.00003 | -0.158 | -0.025 |
| | (0.340) | (0.054) | (0.310) | (0.055) |
| breed_k | 0.003 | -0.004 | 0.060 | 0.012 |
| | (0.053) | (0.009) | (0.050) | (0.009) |
| social_aware | 0.292 | 0.088** | -0.448** | -0.113*** |
| | (0.244) | (0.039) | (0.227) | (0.040) |
| tech_aware | -0.438* | -0.122*** | 0.543** | 0.143*** |
| | (0.255) | (0.040) | (0.249) | (0.043) |
| cons | 0.441 | - | -1.033 | - |
| | (0.991) | - | (0.969) | - |
| Log likelihood | -413.95 | | | |
| Log likelihood (const) | -461.31 | | | |
| Observation | 422 | | | |

[a] Standard errors are shown in parentheses.

[b] *, **, and *** indicate statistical significance at 10%, 5%, and 1%, respectively.

and interest in technology in general (tech_aware) raised the probability of a favorable one. Interest in social issues (social_aware) raised the likelihood of being unfavorable. Male (d_male) respondents generally had a more positive attitude toward genomics than females did, and older (age) participants were more likely to be negative. Note these outcomes are drawn from the Japanese public, and generalizations outside Japan may require further analysis.

## Conclusions

The purpose of this study has been to investigate the public acceptability of genome editing during this time of increased awareness, particularly in light of the sensational announcement of the Chinese twins. Because the twin-baby news was released in late 2018 and received

**Table 6. Estimated Parameters and Marginal Effects, 2019[a,b].**

| | 2019 | | | |
| --- | --- | --- | --- | --- |
| | Negative | | Positive | |
| | Parameter | Marginal Effect | Parameter | Marginal Effect |
| c_breed | 0.575* | 0.065 | 0.623** | 0.068 |
| | (0.327) | (0.056) | (0.314) | (0.050) |
| c_risk | 1.407*** | 0.260*** | 0.022 | -0.092 |
| | (0.399) | (0.066) | (0.497) | (0.078) |
| c_medical | -0.140 | -0.076 | 0.732** | 0.135*** |
| | (0.320) | (0.056) | (0.291) | (0.047) |
| c_embryo | 1.513*** | 0.290*** | -0.122 | -0.124** |
| | (0.306) | (0.049) | (0.356) | (0.055) |
| c_others | -0.061 | -0.128 | 1.711* | 0.298** |
| | (1.261) | (0.213) | (0.889) | (0.135) |
| age | 0.013* | 0.003*** | -0.012 | -0.003** |
| | (0.007) | (0.001) | (0.007) | (0.001) |
| d_male | -0.438** | -0.108*** | 0.385 | 0.096*** |
| | (0.210) | (0.035) | (0.237) | (0.037) |
| d_univ | -0.054 | -0.026 | 0.235 | 0.044 |
| | (0.209) | (0.036) | (0.223) | (0.035) |
| d_bio | 0.316 | 0.032 | 0.387 | 0.045 |
| | (0.230) | (0.040) | (0.251) | (0.040) |
| breed_k | -0.076* | -0.017*** | 0.047 | 0.013* |
| | (0.039) | (0.007) | (0.044) | (0.007) |
| social_aware | 0.279 | 0.067** | -0.230 | -0.058** |
| | (0.183) | (0.031) | (0.186) | (0.029) |
| tech_aware | -0.361* | -0.110*** | 0.634*** | 0.134*** |
| | (0.211) | (0.036) | (0.223) | (0.034) |
| cons | -0.546 | - | -2.763*** | - |
| | (0.829) | - | (0.883) | - |
| Log likelihood | -651.53 | | | |
| Log likelihood (const) | -738.59 | | | |
| Observation | 677 | | | |

[a] Standard errors are shown in parentheses.

[b] *,**, and *** indicate statistical significance at 10%, 5%, and 1%, respectively.

widespread attention in public as well as academic circles, our 2016–2019 survey horizon offered a valuable opportunity to see how public opinion was affected by the media coverage.

Our respondents who had earlier been exposed to the technology's medical applications tended to have approving-views of genome editing, even in 2018 and 2019. On the other hand, insofar as they were exposed to news of its use with human fertilized eggs, they were especially inclined to oppose it. Revelations about the Chinese twins raised public awareness of genome editing technology in general, along with ethical criticism of it when used for the human performance-enhancement purposes.

Our multinomial logit model doesn't ask how any particular types of ethical concerns affect media responses (Tables 2, 4 and 6). And we are unable here to examine specific ethical concerns in more detail, such as, for example, risks of life or compliance of the medical research. We have seen, however, ethical concern itself moved higher in 2018 and 2019, in the midst of

the twins event (Table 3). Further study of the public's ethical concerns will be required, although the science community must do its own part to keep the public informed about critical technologies' purposes and applications.

Finally, our results do indicate that Japanese have avoided confusing or even associating genomics' medical applications with its farm breeding ones. In fact, familiarity with farm breeding brings a more favorable impression of gene editing overall.

## Supporting information

**S1 Text. Questionnaire.**
(DOCX)

**S1 Table. Summary statistics of pre-registered and survey sample.**
(XLSX)

**S2 Table. Awarenesses and their corresponding impressions.**
(DOCX)

**S3 Table. Summary statistics of survey sample: 'Not being aware' and 'Being aware'.**
(XLSX)

**S4 Table. Summary statistics of respondents included in multinomial logit analysis.**
(XLSX)

**S5 Table. Data and definition.**
(XLSX)

## Author Contributions

**Conceptualization:** Yoko Saito, Ryo Ohsawa.

**Data curation:** Daiki Watanabe.

**Formal analysis:** Daiki Watanabe.

**Funding acquisition:** Yoko Saito, Mai Tsuda.

**Investigation:** Daiki Watanabe, Yoko Saito, Mai Tsuda.

**Methodology:** Yoko Saito.

**Project administration:** Yoko Saito, Ryo Ohsawa.

**Supervision:** Yoko Saito, Ryo Ohsawa.

**Validation:** Mai Tsuda, Ryo Ohsawa.

**Writing – original draft:** Yoko Saito.

**Writing – review & editing:** Yoko Saito.

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
