## [Decision Letter · Decision Letter 0]

12 May 2020

PONE-D-20-07652

Increased Awareness and Decreased Acceptance of Genome-Editing Technology: The Impact of Chinese Twin Babies

PLOS ONE

Dear Dr. Saito,

Thank you for submitting your manuscript to PLOS ONE. After careful consideration, we feel that it has merit but does not fully meet PLOS ONE’s publication criteria as it currently stands. Therefore, we invite you to submit a revised version of the manuscript that addresses the points raised during the review process.

The manuscript and the reviewers’ comments were carefully evaluated. The manuscript was appreciated by the Reviewers. Nevertheless, as suggested, the manuscript requires some improvements before to be considered for publication, particularly about the methods description. Further suggested revisions are in detail reported in the Reviewers’ comments.

We would appreciate receiving your revised manuscript by Jun 26 2020 11:59PM. To enhance the reproducibility of your results, we recommend that if applicable you deposit your laboratory protocols in protocols.io, where a protocol can be assigned its own identifier (DOI) such that it can be cited independently in the future. For instructions see: http://journals.plos.org/plosone/s/submission-guidelines#loc-laboratory-protocols

We look forward to receiving your revised manuscript.

Kind regards,

Simone Garzon

Academic Editor

PLOS ONE

2. Please confirm data were analyzed anonymously or alternatively, please provide additional details regarding participant consent. In the ethics statement in the Methods and online submission information, please ensure that you have specified (1) whether consent was informed and (2) what type you obtained (for instance, written or verbal, and if verbal, how it was documented and witnessed). If your study included minors, state whether you obtained consent from parents or guardians. If the need for consent was waived by the ethics committee, please include this information.

Reviewers' comments:

Reviewer's Responses to Questions

**Comments to the Author**

1. Is the manuscript technically sound, and do the data support the conclusions?

Reviewer #1: Yes

Reviewer #2: Yes

Reviewer #3: Partly

2. Has the statistical analysis been performed appropriately and rigorously? 

Reviewer #1: I Don't Know

Reviewer #2: I Don't Know

Reviewer #3: No

3. Have the authors made all data underlying the findings in their manuscript fully available?

Reviewer #1: Yes

Reviewer #2: Yes

Reviewer #3: Yes

4. Is the manuscript presented in an intelligible fashion and written in standard English?

Reviewer #1: Yes

Reviewer #2: Yes

Reviewer #3: Yes

5. Review Comments to the Author

Reviewer #1: Thank you for this manuscript. The article presents the survey clearly and thoroughly. I found it very interesting to read. I would like to see a few more details on where the study was conducted, particularly about the location of the survey respondents. Was the survey distributed globally? What languages were used? Some comments in the article suggest that the survey was conducted in Japan only (e.g. p. 11, line 143). It would also be important to discuss the implications of this. If the survey was conducted in one country only, this needs to be stated and an acknowledgement made that this may limit the generalizability of its findings.

One small formatting issue is that reference numbers were regularly used as subjects in sentences, e..g. “[3] found that their results …” This approach is unfamiliar to me and should be checked against the journal's style guidelines.

Otherwise, this is a very good manuscript.

Reviewer #2: I think this manuscript provides important findings about the impacts of the Chinese human embryo editing controversy on public opinion about gene editing in Japan. As such, I support its publication. However, I have a few suggestions for the authors to consider:

1. I found it confusing that the article starts with the discussion of gene editing of plants and animals ("breeding" applications), even though the primary focus is on human gene editing. I think it would therefore be better to start the paper with a discussion of human gene editing and the Chinese incident. At the end of the current intro, it comes back to use of gene editing for animals and plants, which is appropriate there because one of the interesting secondary findings of the paper is that controversy over Chinese human embryo editing dis not seem to appreciably affect public opinion on breeding applications of gene editing. But it confusing to focus on animal and plant applications in the first paragraph of the paper given that the primary focus and finding is on impact on opinion about medical and embryo use of gene editing.

2. On lines 104-05, the manuscript states that the subjects were randomly selected from a "pre-registration sample." It would be useful to say more about how this pre-registraton sample was created. Was it created from a particular geographical region? How were people solicited for the pre-registration sample?-These details are important for understanding the representativeness of the sample.

3. My understanding is that the subjects were all from Japan. That should be explicitly stated. Also, it would be useful to have a little bit of information about the overall perspective of the Japanese public. For example, Japan is a world leader in stem cells, does this make the public more supportive of embryo science generally, or is it similar to European and the Americas?

4. Table 1 states that there was unequal number of males and females in 2019 (even though text says same numbers each year. How unequal was the 2019 sample, did it favor males or females, and was this likely to have had any effect on the results?

5. In table 3, I did not understand the headings “Most Impressionable” (for 2016) vs “Describes Me” (for 2018 and 2019). The meaning of these terms should be explained better.

6. In the conclusion, the authors state that public requires requires researchers to strictly follow ethical guidelines. This might be a reasonable speculation - but was not directly tested by the study. It may be that people are disturbed by embryo manipulations even if it does follow ethical guidelines. Therefore the role of ethical guidelines should be presentrd as a reasonable hypothesis, or excluded altogether since it was not evaluated in this study.

– what does this mean ?and more males or females?

Reviewer #3: I read with great interest the Manuscript titled “Increased Awareness and Decreased Acceptance of Genome-Editing Technology: The Impact of Chinese Twin Babies” (PONE-D-20-07652).

I was particularly pleased to review this paper. In my honest opinion, the topic is interesting enough to attract the readers’ attention. Methodology seems appropriate and conclusions are supported by the data analysis. Nevertheless, authors should clarify different points of methods and improve the discussion discussing limitations of the study that are not evidenced in the discussion.

In general, the Manuscript may benefit from several major revisions, as suggested below:

- All the text needs a language revision in order to improve some typos and grammatical errors.

- Please, check in all the text the use of references as subject. This is not correct, use the form Surname et al. [reference] verb …

- I would suggest checking the use of abbreviation in the abstract and in the main text, they need to be reported in the extended form at the first use.

- I would suggest checking the authors guidelines for the manuscript format.

- I would suggest providing more information about the target population included in the Macromill database, the geographic area, and how the list of the company has been built. Is it actually representative of the entire population? All these details are required to understand the possible biases and limitations of the survey which need to be better discussed.

- I would suggest providing data about non responders and discuss the possible role of this proportion of subjects on the study results. This can be a source of bias and limitations. Are more characteristics of the investigated population available, such age, education, or other? The same about who did not respond.

- I would suggest providing more detail about “how” the survey was conducted. The modality, if interviewers were adopted, which language, if the survey was validated and the process of validation. In general, more details about the methods are required to understand the actual value of provided results and specifically the inference from these results.

- I would suggest adding a methods section with deep detailed description of survey development with validation, survey distribution, target population (geographic area, language, details about list). The Authors should add this section with the aim to allow the repetition of the study by other investigators, key information and steps used to product the results need to be reported.

6. PLOS authors have the option to publish the peer review history of their article (what does this mean?). If published, this will include your full peer review and any attached files.

Reviewer #1: Yes: Dónal O’Mathúna

Reviewer #2: No

Reviewer #3: No

---

## [Author Response · Author response to Decision Letter 0]

26 Jun 2020

Dear Reviewers.

Your comments are insightful, greatly useful for our revision, and are much appreciated. As you recommend, we have now shifted our focus to the medical application of genome editing technology and generally away from the agricultural and fish breeding aspects. This shift allows us to contribute more widely to the literature on the technological use of genome editing, in both the scientific and general community. 

Please refer to the "Reply to the reviewers' comment" for the further details of our reply.

Thank you again for your invitation to revise.

---

## [Decision Letter · Decision Letter 1]

11 Aug 2020

Increased Awareness and Decreased Acceptance of Genome-Editing Technology: The Impact of the Chinese Twin Babies

PONE-D-20-07652R1

Dear Dr. Saito,

We’re pleased to inform you that your manuscript has been judged scientifically suitable for publication and will be formally accepted for publication once it meets all outstanding technical requirements.

Kind regards,

Simone Garzon

Academic Editor

PLOS ONE

Additional Editor Comments (optional):

Reviewers' comments:

Reviewer's Responses to Questions

**Comments to the Author**

1. If the authors have adequately addressed your comments raised in a previous round of review and you feel that this manuscript is now acceptable for publication, you may indicate that here to bypass the “Comments to the Author” section, enter your conflict of interest statement in the “Confidential to Editor” section, and submit your "Accept" recommendation.

Reviewer #1: All comments have been addressed

Reviewer #3: All comments have been addressed

2. Is the manuscript technically sound, and do the data support the conclusions?

Reviewer #1: Yes

Reviewer #3: Yes

3. Has the statistical analysis been performed appropriately and rigorously? 

Reviewer #1: I Don't Know

Reviewer #3: Yes

4. Have the authors made all data underlying the findings in their manuscript fully available?

Reviewer #1: Yes

Reviewer #3: Yes

5. Is the manuscript presented in an intelligible fashion and written in standard English?

Reviewer #1: Yes

Reviewer #3: Yes

6. Review Comments to the Author

Reviewer #1: Thank you for responding to my earlier comments, particularly making it very clear that this survey was conducted with the Japanese public. While the article is generally written well, it could benefit from one more careful proofreading by someone with excellent English-writing skills. There are places where the English currently is awkward. e.g. p. 2, line 32 "ask" should be "inquire"; p. 3, line 45, the concluding phrase is incomplete grammatically; p. 14, line 187, "the" should be inserted between "was" and "highest".

Otherwise, the article is very good.

Reviewer #3: I carefully evaluated the revised version of this manuscript.

Authors have performed the required changes, improving significantly the quality of the paper.

7. PLOS authors have the option to publish the peer review history of their article (what does this mean?). If published, this will include your full peer review and any attached files.

Reviewer #1: **Yes: **Dónal O'Mathúna

Reviewer #3: No

---

## [Editor Report · Acceptance letter]

8 Sep 2020

PONE-D-20-07652R1 

Increased Awareness and Decreased Acceptance of Genome-Editing Technology: The Impact of the Chinese Twin Babies 

Dear Dr. Saito:

I'm pleased to inform you that your manuscript has been deemed suitable for publication in PLOS ONE. Congratulations! Your manuscript is now with our production department. 

Kind regards, 

on behalf of

Dr. Simone Garzon 

Academic Editor

PLOS ONE